# Head-to-Head Comparison of Two Nomograms Predicting Probability of Lymph Node Invasion in Prostate Cancer and the Therapeutic Impact of Higher Nomogram Threshold

**DOI:** 10.3390/jcm10050999

**Published:** 2021-03-02

**Authors:** Zilvinas Venclovas, Tim Muilwijk, Aivaras J. Matjosaitis, Mindaugas Jievaltas, Steven Joniau, Daimantas Milonas

**Affiliations:** 1Department of Urology, Lithuanian University of Health Sciences, Medical Academy, LT-44307 Kaunas, Lithuania; AivarasJonas.Matjosaitis@lsmuni.lt (A.J.M.); Mindaugas.Jievaltas@lsmuni.lt (M.J.); Daimantas.Milonas@lsmuni.lt (D.M.); 2Department of Urology, Leuven University Hospital, 3000 Leuven, Belgium; tim.muilwijk@kuleuven.be (T.M.); steven.joniau@kuleuven.be (S.J.)

**Keywords:** prostate cancer, radical prostatectomy, pelvic lymph node dissection, lymph node invasion, preoperative nomogram

## Abstract

*Introduction*: The aim of the study was to compare the performance of the 2012 Briganti and Memorial Sloan Kettering Cancer Center (MSKCC) nomograms as a predictor for pelvic lymph node invasion (LNI) in men who underwent radical prostatectomy (RP) with pelvic lymph node dissection (PLND), to examine their performance and to analyse the therapeutic impact of using 7% nomogram cut-off. *Materials and Methods*: The study cohort consisted of 807 men with clinically localised prostate cancer (PCa) who underwent open RP with PLND between 2001 and 2019. The area under the curve (AUC) of the receiver operator characteristic analysis was used to quantify the accuracy of the 2012 Briganti and MSKCC nomograms to predict LNI. Calibration plots were used to visualise over or underestimation by the models and a decision curve analysis (DCA) was performed to evaluate the net benefit associated with the used nomograms. *Results*: A total of 97 of 807 patients had LNI (12%). The AUC of 2012 Briganti and MSKCC nomogram was 80.6 and 79.2, respectively. For the Briganti nomogram using the cut-off value of 7% would lead to reduce PLND in 47% (379/807), while missing 3.96% (15/379) cases with LNI. For the MSKCC nomogram using the cut-off value of 7% a PLND would be omitted in 44.5% (359/807), while missing 3.62% (13/359) of cases with LNI. *Conclusions*: Both analysed nomograms demonstrated high accuracy for prediction of LNI. Using a 7% nomogram cut-off would allow the avoidance up to 47% of PLNDs, while missing less than 4% of patients with LNI.

## 1. Introduction

Prostate cancer (PCa) is the second most frequently diagnosed cancer among males worldwide and one of the most used treatment options for localised or locally advanced PCa is radical prostatectomy (RP) [1,2]. Pelvic lymph node dissection (PLND) is performed on patients with intermediate or high-risk PCa but it should be omitted for patient with low risk PCa [2,3]. PLND represents the important staging procedure in identifying patients with lymph node invasion (LNI); therefore, it is related to poor prognosis after RP. The rate of LNI increases linearly with the number of removed LN, which is why more extended pelvic lymph node dissection (ePLND) is recommended [4]. However, ePLND is associated with increased morbidity, longer operating time and higher costs; for these reasons, it should be done with high precaution [5]. Instrumental tests such as computer tomography, choline PET/CT or magnetic resonance imaging does not reach clinically acceptable diagnostic accuracy for detection of LNI and may show false-negative results [6]. Only prostate-specific membrane antigen-based PET/CT has higher sensitivity for LNI. However, smaller than 4 mm metastases are likely to be missed [7]. Aiming to identify patient, for whom PLND could be omitted, several original and updated nomograms predicting the risk of LNI have been proposed [8,9,10,11,12,13,14,15]. Nonetheless, only some of them are currently used because each predictive model should undergo external validation before introducing it into wide practice [9,11,13,14]. Choosing the optimal nomogram cut-off is essential for differentiating patients for PLND.

According to the European Association of Urology guidelines, PLND should be performed for patients when the predicted probability of LNI surpasses 5% [2]. However, in a few recent reports 7% was suggested as an optimal cut-off with a similar sensitivity and specificity and higher number of patients for whom PLND could be safely omitted [10,16,17].

The aim of our study was to compare the performance of the two most used MSKCC and 2012 Briganti preoperative risk nomograms for prediction of LNI in a cohort of men undergoing open radical prostatectomy and to assess the accuracy of 7% cut-off for performing PLND.

## 2. Patients and Methods

The study cohort consisted of 807 men with clinically localised PCa who underwent open RP with PLND between January 2001 and December 2019 at the Lithuanian University of Health Sciences, department of Urology. Pathologic analysis of the pre- and post-operative specimens was performed by dedicated uropathologists. Descriptive measurements included preoperative clinical and biopsy data: age, clinical stage (cT), prostate specific antigen (PSA), primary and secondary biopsy Gleason pattern and percentage of positive cores (defined as the number of positive cores over the total number of cores taken). After the surgery the post-operative Gleason score, pathological stage (pT), lymph node status (LNI), number of lymph nodes removed and the number of positive lymph nodes were registered. Pathological stage was assessed using the 2002 TNM system and tumour grading was classified using the revised 2005 Gleason grading system [18] and a new grading system for patients who underwent RP after 2014 [19].

The PLND template was changed during the study period. Up to 2012, limited PLND (lPLND) removing fatty tissue located within the obturator fossa was the most common procedure. Since 2012, the ePLND template has been adapted and involves removal of nodes overlying the external iliac vessels and internal iliac artery and obturator fossa. As an option, areas of the common iliac artery and the presacral region can also be included. Because of retrospective design, we have no data which template of dissection was chosen in every case. The number of LNs examined served as a surrogate of the extent of the PLND. All men were divided into two groups according to the number of removed lymph nodes: <10 LNs (lPLND) vs. ≥10LNs (ePLND).

Biochemical recurrence (BCR) was defined as two consecutive PSA values ≥ 0.2 ng/mL. Clinical progression (CP) was identified when skeletal or visceral lesions were confirmed by bone scan, computer tomography (CT), positron emission tomography (PET/CT), or magnetic resonance imaging (MRI); local or loco-regional recurrence was confirmed by biopsy, salvage surgery or MRI/CT. Biochemical progression free survival (BPFS) was defined as the time from the operation to the day of BCR, clinical progression free survival (CPFS) was defined as the time from the operation to the day of CP and cancer-specific survival (CSS) was defined as the time from the operation to the day of death from PCa.

The Lithuanian University of Health Sciences Ethical Committee approved prospective collection of the data (BE-2-48) and all patients signed a consent form provided before the RP. This study follows the rules of the Declaration of Helsinki.

## 3. Statistical Aanalysis

Medians, interquartile ranges and frequencies were used for descriptive statistics. Chi-square and *t*-tests were used to compare difference in medians between lymph node positive vs. lymph node negative patients.

The data set was used to compare two different tools predicting the probability of LNI in patients with prostate cancer. These were the 2012 Briganti nomogram [20] and MSKCC nomogram [21]. Within the Briganti nomogram, the predictor variables consist of the preoperative PSA level, clinical stage, biopsy Gleason sum and the percentage of positive biopsy score. Within the MSKCC nomogram, the predictor variables consist of the patient age, preoperative PSA level, clinical stage, biopsy Gleason sum and the percentage of the positive biopsy score. The area under the curve (AUC) of the receiver operator characteristic (ROC) analysis was used to quantify the accuracy of the different models to predict LNI. The specificity, sensitivity, and negative predictive value (NPV) were calculated for each nomogram-derived LNI probability cut-off. Moreover, the extent of over and under estimation of the observed LNI rate was created graphically in logistic calibration plots and a decision curve analysis (DCA) was performed to evaluate the net benefit associated with the used nomograms.

The template of PLND has been changed from limited at the beginning of the study to a more extended. The lPLND might be associated with lower detection rate of LNI and increased the risk of false negative results. Therefore, we performed a sub analysis. All patients with LNI were excluded. In order to ensure that false negative results would not give any impact to our findings, BPFS, CPFS, CSS were estimated using Kaplan-Meier curves and log-rank test was used to compare difference in survival between lPLND vs. ePLND groups in LN negative cohort.

## 4. Results

The characteristics of the 807 patients are listed in the Table 1. The median (interquartile) number of removed lymph nodes was 7 [4,5,6,7,8,9,10,11]. The overall rate of LNI in the entire cohort was 12% (97/807). 43 patients (44.3%) had one positive lymph node, 27 patients (27.85%) had two and 27 patients (27.85%) had three or more positive lymph nodes. Median preoperative PSA levels were 12.4 ng/mL (8.3–19.9) versus 10.1 ng/mL (6.3–14.2) for patients with and without LNI, respectively (*p* < 0.001). Men with LNI showed higher rates of cT3 (61.8% vs. 23.3%; *p* < 0.001) as well as higher rates of biopsy Gleason score ≥ 8 (42.3% vs. 15.3%; *p* < 0.001), median percentage of positive biopsy cores (62% vs. 40%; *p* < 0.001), higher rates of pathological Gleason score ≥ 8 (66% vs. 18.1%; *p* < 0.001), pathological stage ≥ pT3 (93.8% vs. 50.7%; *p* < 0.001) and positive surgical margin (61.9% vs. 37.6%; *p* < 0.001).

The accuracy of the 2012 Briganti nomogram for prediction of LNI in the study cohort using ROC analysis was 80.6 (95% Cl 75.9–85.4); as for the MSKCC nomogram the accuracy was 79.8 (95% Cl 75.2–84.5) (Figure 1).

Figure 2 shows the graphic comparison of the nomogram-predicted probabilities and the actual fraction of LNI within the cohort. Both calibration plots are quite similar, but there is a slightly better calibration of the 2012 Briganti nomogram in higher predicted risk > 50%.

The DCA demonstrated that both nomograms improved clinical risk prediction against threshold probabilities of LNI ≤ 20% (Figure 3).

Additionally, we calculated the accuracy of various possible cut-offs for the prediction of LNI using both nomograms. For the 2012 Briganti nomogram using the cut-off value of 7% (NPV 96.04%, sensitivity 84.54% and specificity 51.27%) would lead to reduce PLND in 47% (379/807), while missing 3.96% (15/379) cases with LNI (Table 2). For the MSKCC nomogram using the cut-off value of 7% (NPV 96.38%, sensitivity 86.6% and specificity 48.73%) a PLND would be omitted in 44.5% (359/807), while missing 3.62% (13/359) of cases with LNI (Table 3).

Furthermore, our sub analysis demonstrates that the majority of the patients underwent lPLND (558/807; 69.1%). Men who underwent ePLND showed higher rates of median preoperative PSA as well as higher rates of cT3, biopsy Gleason score ≥ 8, pathological stage ≥ pT3, pathological Gleason score ≥ 8, LNI and number of positive ≥ 3 LNs. However, there were no significand difference between the positive surgical margin, BCR, CP and cancer related death (Appendix A
Appendix A).

After excluding the patients with LNI, long-term oncological outcomes were analysed. Median time of follow-up after RP was 82.5 (IQR 39–136) months. Over this time, 204 out of 710 men (28.7%) experienced BCR. CP was diagnosed in 39 (5.5%) cases. During the follow-up, 92 patients (13.0%) died. In 19 cases (2.7%) PCa was the cause of death.

Among patients with pN0, the estimated 10-year BPFS rate differed comparing limited vs. extended PLND subgroups (61.3% vs. 37.8%; *p* < 0.001); However, no difference was detected when comparing CPFS and CSS in these subgroups (88.5% vs. 78.3%; *p* = 0.2 and 95.0% vs. 98.0% *p* = 0.7, respectively) (Appendix A).

## 5. Discussion

PLND during RP remains the most accurate staging procedure for the detection of LNI in PCa [8]. While performing extended PLND, more than 90% of patients would be staged correctly [4,22]. However, the therapeutic effect of PLND remains questionable. There have been few studies demonstrating therapeutic benefit of PLND [23,24]. However, a recent systematic review concludes that PLND is associated with worse intraoperative, perioperative outcomes and failed to improve oncological outcomes [5]. Therefore, proper selection of men for this procedure is needed to avoid it’s unnecessarily performance. Several original and updated nomograms which predicted the risk of LNI have been proposed [8,9,10,11,12,13,14]. The EAU-EANM-ESTRO-ESUR-SIOG and NCCN guidelines recommend the use of MSKCC or Briganti nomograms as a first line option to determine whether the patient should be considered for an PLND [2,25]. It should be noted that there are some requirements for nomograms: the tools should be periodically updated, it should be possible to use it in daily practice and it should be externally validated on a large intercontinental patient cohort to determine its accuracy. For example 2012 Briganti nomogram is being used more often compared to the improved 2017 Briganti nomogram version, because it requires including more variables in the main formula (percentage of cores with high-grade disease and percentage of cores with lower-grade disease) [10,26]. As R. Turo et al. highlighted, genetic and phenotypic differences in tumour biology and different PSA assays are relevant factors in different cohorts of patients and may play a huge role in nomogram accuracy [27]. For that reason, several head-to-head comparisons of preoperative nomograms of different patient cohorts have been made. M. Bandini et al. [26] compared four different nomograms: Cagiannos, Godoy, the 2012 Briganti and the online-MSKCC nomograms.

Despite several comprehensive analytical steps, they did not prove that one nomogram is superior to another. Furthermore, all nomograms achieved the same accuracy for predicting LNI and their ability to avoid unnecessary PLND was quite equal. Hueting et al. [28] with a cohort that consisted of 1001 men with the LNI rate 28% externally validated 16 prediction models. They found that the 2012 Briganti and MSKCC nomograms showed the highest AUC of 0.76 and 0.75, respectively. However, these results did not match with Hinev et al. [29] findings which surprisingly showed that the 2012 Briganti nomogram was far superior to the MSKCC nomogram, reporting a calculated AUC of 0.875 vs. 0.77.

In our presented cohort both predictive tools demonstrated similar net benefit at DCA. The 2012 Briganti nomogram achieved an AUC of 80.6 compared to MSKCC nomogram that achieved an AUC of 79.8 with prevalence of LNI at 12%.

Several possible cut-offs have been suggested from which a PLND could be omitted [10,16,17,30]. According to the EAU guidelines, PLND should be suggested for patient having intermittent or high risk PCa features and when the predicted probability of LNI surpasses 5% [2]. After a detailed analysis we demonstrated, that the cut-off with the best trade-off between PLND omission and missing LNI for the 2012 Briganti and MSKCC nomograms was 7%. In the original series, Briganti et al. [15] recommend to use a 5% cut-off that would allow avoiding up to 65% PLND while missing only 1.5% patients with LNI. In our cohort, the optimal cut-off for the Briganti nomogram was a bit higher (7%) and it would lead to reduce PLND in 47% while missing 3.96% cases with LNI. We should point out several differences that have become evident comparing our data sets with those presented by Briganti et al. The first one is the descriptive characteristic of the patients, with the difference evident in: the comparison of median PSA (10.3 ng/mL vs. 6.3 ng/mL), frequency of clinical stage cT3 (27.9% vs. 5.3%), median percentage of positive biopsy cores (50.0% vs. 35.5%), pathological Gleason score ≥ 8 (23.8% vs. 10.7%) and LNI (12% vs. 8.3%). Furthermore, Briganti et al. analysed a cohort where more than 50% of men underwent PLND with a cut-off at 3%, while in our presented study such patients accounted for 25% of our cohort. These findings demonstrated that the patent characteristic in the Briganti series consisted of lower PCa aggressiveness. Similar differences in cancer characteristics were found when we compared our MSKCC nomogram-derived cut-off results with research by Godoy et al. [14]. This demonstrated that the RP performed for men with more aggressive cancer is associated with lower number of omitted PLND. However, sensitivity, specificity and NPV of suggested cut-off in our analysis and original reports are similar. Thus, our findings are in concordance with previous reports which suggested a 7% cut-off as optimal for PLND [10,16,17].

Taken together, our analysis demonstrated high prediction accuracy of both the 2012 Briganti and MSKCC nomograms for predicting LNI. Suggested 7% cut-off of these nomograms could be used as an acceptable tool for differentiating patients for PLND with higher specificity, similar sensitivity and NPV when compared to 5% cut-off. Moreover, the 7% threshold would increase the number of patients for whom PLND could be safely omitted by up to 10% when compared with the currently used 5% cut-off.

The retrospective nature of our study represents some limitation, which may have influenced our results. During the study period the median number of cores taken during biopsy has increased up to 10 and sextant biopsy is no longer considered adequate. It should be noted that at the beginning of the study, only few patients underwent pelvic magnetic resonance imaging (MRI) before prostate biopsy, which is used in majority of cases in today’s practice. MRI findings improve the detection of cancer with more aggressive features [2] and may change the percentage of positive cores that could lead to increase in the calculated probability of lymph node involvement.

Moreover, changes in the Gleason score after the 2014 International Society of Urological Pathology recommendations could have influenced interpretation of cancer aggressiveness as well as the calculated risk probability of LNI.

Finally, variability in surgical technique might have contributed to differences in the number of lymph nodes removed, since the template of PLND has been changed from limited at the beginning of the study to a more extended. This might be associated with lower detection rate of LNI and the risk of false negative results at final pathology. For that reason, we performed a sub analysis where we divided patients according to the number of removed lymph nodes: from one to nine LNs (lPLND) vs. ≥10 LNs (ePLND). Patients who underwent ePLND had more cases of LNI as well as more aggressive cancer features. Other authors have also pointed out that the rate of LNI increases linearly with the number of removed LN and it depends on the aggressiveness of PCa [4,24,31,32]. The main concern with our analysis is that by insufficiently performing PLND could lead in missing positive nodes which in turn might influence our results. Hence, we evaluated the long-term survival analysis in node-negative patients. Surprisingly, patients who underwent ePLND had lower BPFS rate. However, in our opinion, it was more associated with worse PCa features than with the number of removed LNs. Interestingly, the extend of PLND did not appear to have any effect on CPFS and CSS.

Such findings demonstrate that although the number of removed LNs in our cohort was not high achieved results could be used in clinical practice.

## 6. Conclusions

We compared the 2012 Briganti and MSKCC nomograms—the two most often used nomograms for prediction of lymph node invasion in men undergoing pelvic lymph node dissection at radical prostatectomy. Both of them demonstrated high accuracy with similar net benefit at DCA and can be used in daily practice. Our results showed that a 7% nomogram cut-off would allow the avoidance up to 47% of PLNDs, while missing less than 4% of patients with LNI.

## Figures and Tables

**Figure 1 jcm-10-00999-f001:**
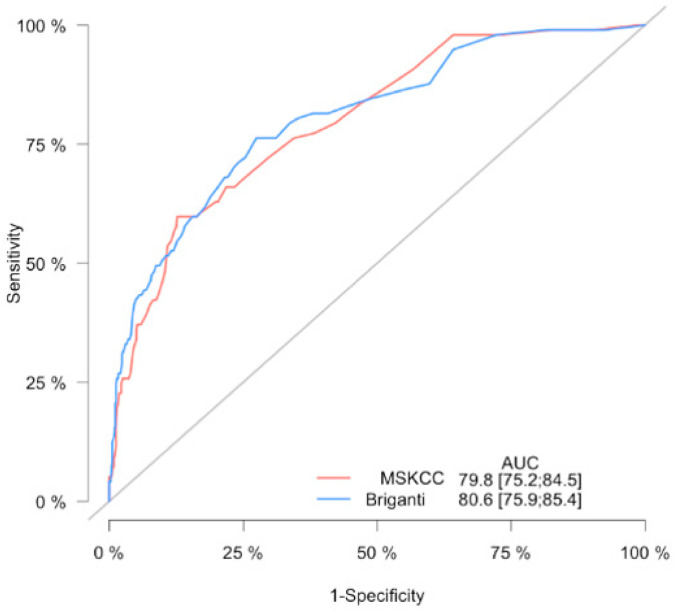
Receiving operator characteristic curve for the lymph node invasion prediction model. The area under the curve (AUC) for 2012 Briganti and MSKCC nomograms.

**Figure 2 jcm-10-00999-f002:**
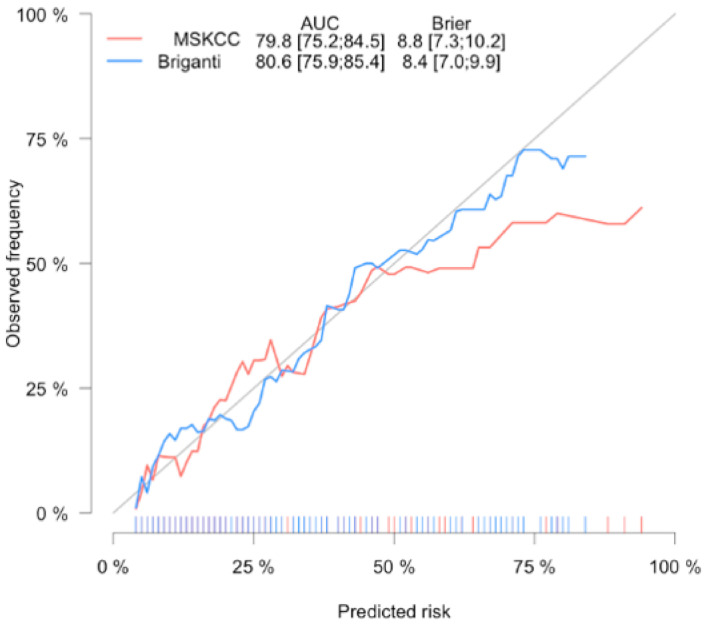
Calibration plot of observed proportion versus predicted probability of lymph node invasion of the 2012 Briganti and MSKCC nomograms.

**Figure 3 jcm-10-00999-f003:**
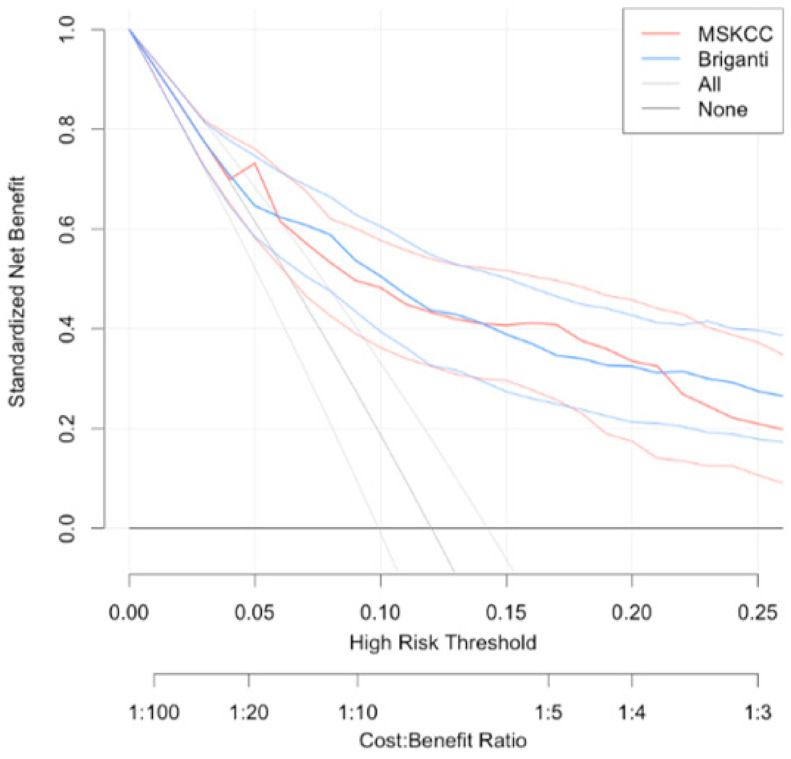
Decision curve analysis demonstrating the net benefit associated with the use of 2012 Briganti (blue line) MSKCC (red line) nomograms for the detection of lymph node invasion.

**Table 1 jcm-10-00999-t001:** Descriptive characteristics of the study cohort.

Parameter	pN0 (*n* = 710)	pN1 (*n* = 97)	*p* Value	All (*n* = 807)
Age (yr): median, (IQR)	65 (60–69)	64 (57–67.5)	0.034	65 (60–69)
PSA (ng/mL): median, (IQR)	10.1 (6.28–14.23)	12.4 (8.27–19.85)	<0.001	10.3 (6.5–14.7)
Clinical stage: *n*, (%)			<0.001	
cT1	101 (14.2)	3 (3.1)		104 (12.9)
cT2	444 (62.5)	34 (35.1)		478 (59.2)
cT3	165 (23.3)	60 (61.8)		225 (27.9)
Biopsy Gleason Score: n, (%)			<0.001	
6	286 (40.3)	11 (11.3)		297 (36.8)
3 + 4	258 (36.3)	25 (25.8)		283 (35.1)
4 + 3	57 (8.0)	20 (20.6)		77 (9.5)
8	76 (10.7)	22 (22.7)		98 (12.1)
9–10	33 (4.6)	19 (19.6)		52 (6.4)
% of positive cores: median, (IQR)	40 (25–62)	62 (38–88)	<0.001	50 (25–64.5)
Pathological Gleason Score: n, (%)			<0.001	
6	136 (19.2)	1 (1.0)		137 (16.9)
3 + 4	334 (47.0)	13 (13.4)		347 (43)
4 + 3	112 (15.8)	19 (19.6)		131 (16.2)
8	60 (8.5)	15 (15.5)		75 (9.3)
9–10	68 (9.6)	49 (50.5)		117 (14.5)
Pathologic stage: n, (%)			<0.001	
pT2	350 (49.3)	6 (6.2)		356 (44.3)
pT3a	278 (39.2)	29 (29.9)		307 (38)
pT3b	82 (11.5)	60 (61.9)		142 (17.6)
pT4	0	2 (2.1)		2 (0.2)
No, of LN removed: median, (IQR)	6 (4–10)	11 (8–18)	<0.001	7 (4–11)
Positive surgical margin: *n*, (%)	267 (37.6)	60 (61.9)	<0.001	327 (40.5)
MSKCC: median, (IQR)	8 (4–16)	32 (12–48.5)	<0.001	9 (5–20)
Briganti: median, (IQR)	7 (3–17)	37 (16–67.5)	<0.001	8 (4–22)

IQR–interquartiles range, N0–negative lymph node, N1–positive lymph node, PSA–prostate specific antigen, LN–lymph nodes, MSKCC–Memorial Sloane Kettering Cancer Center nomogram.

**Table 2 jcm-10-00999-t002:** Analysis of the 2012 Briganti nomogram-derived cut-offs used to differentiate between men with or without lymph node invasion.

Cut-off	Patients in Whom PLND is Not Recommended According to the Cut-Off (below Cut-Off)	Missing %	Patients below Cut-Off without Histologic LNI	Patients below Cut-Off with Histologic LNI	Patients above Cut-Off without Histologic LNI	Patients above Cut-Off with Histologic LNI	NPV
1	54 (6.7)	0	54 (7.61)	0 (0)	656 (92.39)	97 (100)	100
2	132 (16.4)	0.76	131 (18.45)	1 (1.03)	579 (81.55)	96 (98.97)	99.24
3	199 (24.7)	1	197 (27.75)	2 (2.06)	513 (72.25)	95 (97.94)	99
4	259 (32.1)	1.93	254 (35.77)	5 (5.15)	456 (64.23)	92 (94.85)	98.07
5	298 (36.9)	4.03	286 (40.28)	12 (12.37)	424 (59.72)	85 (87.63)	95.97
6	328 (40.6)	3.96	315 (44.37)	13 (13.4)	395 (55.63)	84 (86.6)	96.04
7	379 (47)	3.96	364 (51.27)	15 (15.46)	346 (48.73)	82 (84.54)	96.04
8	420 (52)	4.05	403 (56.76)	17 (17.53)	307 (43.24)	80 (82.47)	95.95
9	438 (54.3)	4.11	420 (59.15)	18 (18.56)	290 (40.85)	79 (81.44)	95.89
10	458 (56.8)	3.93	440 (61.97)	18 (18.56)	270 (38.03)	79 (81.44)	96.07
15	538 (66.7)	4.28	515 (72.54)	23 (23.71)	195 (27.46)	74 (76.29)	95.72

**Table 3 jcm-10-00999-t003:** Analysis of the MSKCC nomogram-derived cut-offs used to differentiate between men with or without lymph node invasion.

Cut-off	Patients in whom PLND Is not Recommended According to the cut-off (below cut-off)	Missing %	Patients below Cut-off without Histologic LNI	Patients below Cut-off with Histologic LNI	Patients above Cut-off without Histologic LNI	Patients above Cut-off with Histologic LNI	NPV
1	10 (1.2)	0	10 (1.4)	0 (0)	700 (98.6)	97 (100)	100
2	69 (8.6)	1.4	68 (9.6)	1 (1.03)	642 (90.4)	96 (98.97)	98.6
3	123 (15.2)	0.8	122 (17.2)	1 (1.03)	588 (82.8)	96 (98.97)	99.2
4	190 (23.5)	1.05	188 (24.48)	2 (2.06)	522 (73.52)	95 (97.94)	98.95
5	256 (31.7)	0.78	254 (35.77)	2 (2.06)	456 (64.22)	95 (97.94)	99.22
6	316 (39.2)	2.85	307 (43.24)	9 (9.28)	403 (56.76)	88 (90.72)	97.15
7	359 (44.5)	3.62	346 (48.73)	13 (13.4)	364 (51.27)	84 (86.6)	96.38
8	393 (48.7)	4.07	377 (53.1)	16 (16.49)	333 (46.9)	81 (83.51)	95.93
9	431 (53.4)	4.64	411 (57.89)	20 (20.62)	299 (42.11)	77 (79.38)	95.36
10	460 (57)	4.78	438 (61.69)	22 (22.68)	272 (38.31)	75 (77.32)	95.22
15	561 (69.5)	5.53	530 (74.65)	31 (31.96)	180 (25.35)	66 (68.04)	94.47

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
