# Peer review of "Head-to-Head Comparison of Two Nomograms Predicting Probability of Lymph Node Invasion in Prostate Cancer and the Therapeutic Impact of Higher Nomogram Threshold"

_jcm, 2021, doi:10.3390/jcm10050999_

Round 1

Reviewer 1 Report

Review jcm-1065933:"Validation and head-to-head comparison of two nomograms predicting probability of lymph node invasion in prostate cancer and the therapeutic impact of higher nomogram threshold"

Manuscript is well written, the discussion is interesting, and the literature is up-to-date and well-chosen. Also, the statistical calculations are correctly used.

Unfortunately, the work does not bring anything new and significant. The results obtained by the authors confirm the facts presented in the previous works.

The biggest disadvantages are in the material:
- the authors do not provide the anatomical landmarks of lymphadenectomy in the studied group.
- the number of lymph nodes removed is low - Median for whole group was only 7 LN's.
- in the pN0 group, only 6 LN's (4-10) on average were removed - which proves a very limited anatomical range
- to reliably assess LVI, all patients should have the same template of lymphadenectomy - preferably extended.
How many pN0 patients actually had nodal metastases with a limited extent?
What about the so-called skip laesions?
What is the true nodal staging in the analyzed group?

Above-mentioned flaws are serious and reflects results.

Author Response

Thank you very much for your comments and suggestions. Yes, we know that the biggest disadvantage of our work is a small number of removed lymph nodes (median- 7 LNs). However, this is a retrospective study. The PLND template was changed during the study period. Up to 2012, lPLND removing fatty tissue located within obturator fossa was the most common procedure. Since 2012, the ePLND template has been adapted and involves removal of nodes overlying the external iliac vessels and internal iliac artery and obturator fossa. As an option, areas of the common iliac artery and the presacral region can also be included.

After your commends we performed a sub analysis. All men were divided into two groups according to the number of removed lymph nodes: <10 LNs (lPLND) vs. ≥10LNs (ePLND). Medians and frequencies were used for descriptive statistics. The chi-square and t-tests were used to compare pre- and postoperative characteristics between the following groups.

 Furthermore, from this sub analysis all patients with LNI were excluded in order to ensure that false negative results would not give any impact to our findings and biochemical progression free survival (BPFS), clinical progression free survival (CPFS), cancer specific survival (CSS) rates were estimated using Kaplan-Meier analysis. Surprisingly, patients who underwent ePLND had lower BPFS rate. However, in our opinion it was more associated with worse PCa features than with the number of removed LNs. Interestingly, the extend of PLND did not appear to have any effect of CPFS and CSS.

Such findings demonstrate that although the number of removed LNs in our cohort was not high achieved results could be used in clinical practise.

We have made changes in the main manuscript according to your observations and highlighted it in different colour.

Reviewer 2 Report

In the current paper Authors aimed to perform external validation by comparing the performance of the MSKCC and 2012 Briganti preoperative risk nomograms for prediction of lymph node invasion in a cohort of men undergoing open radical prostatectomy and to assess the accuracy of 7% cut-off for performing PLND.

Authors conclude that both nomograms have good accuracy at decision curve analysis, although a higher cut off of 7% would allow the avoidance up to 47% of PLNDs, while missing less than 4% of patients with lymph node invasion.

Authors should be commended for their efforts. Manuscript is well written, statistical analysis is appropriate and references are accurate. However, Authors should be cautious when proposing a higher cut off for lymph node dissection on the basis of their result, since several intrinsic, methodological limitations might have undermined trustworthiness of reported findings.

Main concern can be ascribed to a relatively low median number of lymph nodes removed (7 lymph nodes removed for the entire cohort). This could have been associated with lower detection rate of lymph node positivity and the non-negligible risk of false negative results at final pathology. Similarly, extension and template of lymph node dissection is not reported.

In this regard, it could be of great clinical interest to add some follow up data, to appraise the percentage of patients showing disease recurrence at the level of loco-regional lymph nodes after radical prostatectomy. As such, we could speculate those patients recurring at the level of regional lymph nodes might have already had micro- or clearly metastatic disease at the time of radical prostatectomy, thus corroborating the hypothesis of a not sufficiently extended lymph node dissection.

Author Response

Thank you very much for your comments and suggestions. The biggest disadvantage of our work is a small number of removed lymph nodes (median- 7 LNs). However, this is a retrospective study. The PLND template was changed during the study period. Up to 2012, lPLND removing fatty tissue located within obturator fossa was the most common procedure. Since 2012, the ePLND template has been adapted and involves removal of nodes overlying the external iliac vessels and internal iliac artery and obturator fossa. As an option, areas of the common iliac artery and the presacral region can also be included.

After your commends we performed a sub analysis. All men were divided into two groups according to the number of removed lymph nodes: <10 LNs (lPLND) vs. ≥10LNs (ePLND). Medians and frequencies were used for descriptive statistics. The chi-square and t-tests were used to compare pre- and postoperative characteristics between the following groups.

 Furthermore, from this sub analysis all patients with LNI were excluded in order to ensure that false negative results would not give any impact to our findings and biochemical progression free survival (BPFS), clinical progression free survival (CPFS), cancer specific survival (CSS) rates were estimated using Kaplan-Meier analysis. Surprisingly, patients who underwent ePLND had lower BPFS rate. However, in our opinion it was more associated with worse PCa features than with the number of removed LNs. Interestingly, the extend of PLND did not appear to have any effect of CPFS and CSS.

Such findings demonstrate that although the number of removed LNs in our cohort was not high achieved results could be used in clinical practise.

We have made changes in the main manuscript according to your observations and highlighted it in different colour.

Round 2

Reviewer 1 Report

The efforts of the authors should be appreciated, but the changes introduced only blur the main topic of the work.  

Despite inclusion of additional analyzes,  the main methodological objection regarding the scope of lymphadenectomy still remains - the assumption of the study is to validate the nomograms, which should be performed under homogeneous conditions.

The work should be done on the basis of a unified template of lymphadenectomy - either limited or extended (preferred) and this is where the changes are expected.

Since a majority of the patients underwent lPLND (558/807; 69.1%), maybe the validation of nomograms should apply exactly to this population.

The division of limited (lPLND) vs extended (ePLND) lymphadenectomy introduced by the authors in the additional analysis is somewhat misleading, because in the "Patients and methods" part it refers to the anatomical scope, while in the "Statistical analysis" section it refers to the number of removed nodes and not to the anatomical template. This suggests that in the case of LPND there will always be> 10 LN's removed and in ePLND> 9 LN's which, as we know, is not a rule due to the individual anatomy of the lymphatic system in each patient. 

In addition, questions about new changes to the text:
In each case, was local or loco-regional recurrence confirmed by biopsy or salvage surgery? - it follows from the text.
Also in the newly added sections there are grammatical errors i.e. practise instead practice

Interestingly, the performed subanalysis, its results and conclusions can actually serve as a basis for a separate paper concerning of lymphadenectomy and its impact on oncological results, which has not been resolved to this day. 

Author Response

We agree that heterogeneity of cohort according to the number of removed lymph nodes could be the main weakness of our analysis. However, it is a real life and not always a number of removed lymph nodes is as high as we suspected and extent of dissection as wide as recommended. Moreover, the technique and template of PLND has been changed during the last 20 years. We believe, that validation of nomogram in such cohort is important because it reflects our realistic daily practice and could be useful for majority of centres not only for centres of excellence that presented very high number of removed lymph nodes. Furthermore, the additional survival analyses confirm that the risk of missing positive lymph nodes during the limited dissection is minimal.

Implication of the results in clinical practice from analysis of patients who underwent only limited dissection would be doubtful as general recommendation to perform extended dissection is strong.

Because of retrospective design, we have no data which template of dissection was chosen in every case. For this reason, all cases with the number of removed lymph nodes up to 10 were assigned to limited and 10 and more – to extended dissection. Such classification is not new and used in numerous of published reports [1-4].

We should clarify that local or loco-regional recurrence was confirmed by biopsy or salvage surgery or pelvic MRT/CT.

We agree that survival analysis of subsets with different number of lymph nodes removed could be interesting and it is in our further plans.

Once again, we would like to thank you for reviewing our manuscript and we hope that the last manuscript version is acceptable for you journal and will be interesting for readers.

Sincerely yours,

Zilvinas Venclovas and Daimantas Milonas

  1. Briganti A, Larcher A, Abdollah F, Capitanio U, Gallina A, Suardi N, et al. Updated Nomogram Predicting Lymph Node Invasion in Patients with Prostate Cancer Undergoing Extended Pelvic Lymph Node Dissection: The Essential Importance of Percentage of Positive Cores. European Urology. 2011;61(3):480-7
  2. Hueting TA, Cornel EB, Somford DM, Jansen H, van Basten JA, Pleijhuis RG, et al. External Validation of Models Predicting the Probability of Lymph Node Involvement in Prostate Cancer Patients. Eur Urol Oncol. 2018 10;1(5):411-7.
  3. Hansen J, Rink M, Bianchi M, Kluth LA, Tian Z, Ahyai SA, et al. External validation of the updated briganti nomogram to predict lymph node invasion in prostate cancer patients undergoing extended lymph node dissection. The Prostate. 2013 Jan;73(2):211-8.
  4. Schiavina R, Bertaccini A, Franceschelli A, Manferrari F, Vagnoni V, Borghesi M, et al. The impact of the extent of lymph-node dissection on biochemical relapse after radical prostatectomy in node-negative patients. Anticancer research. 2010 Jun;30(6):2297.

Reviewer 2 Report

In the revised version of the manuscript, Authors significantly improved the overall quality of the paper.

Although some concerns remain according to the limited median number of lymph nodes removed, Authors should be commended for the efforts and for having provided additional analysis in an attempt to answer all doubts raised.

The final result is a well-structured paper

Author Response

Once again, we would like to thank you for reviewing our manuscript and for positive comment of our analyses.

Sincerely yours,

Zilvinas Venclovas and Daimantas Milonas

Round 3

Reviewer 1 Report

I really appreciate the author's comment but ... The authors wrote, quote: "Implication of the results in clinical practice from analysis of patients who underwent only limited dissection would be doubtful as general recommendation to perform extended dissection is strong."

In this case, will the results based on an analysis in which 70% constitutes the ILND be questionable in the authors' opinion?

Maybe the lLND analysis is more likely to reflect the "real life"?

The heterogeneity of the lymphadenectomy template of the analyzed group is the basic objection to the validation process that should be met [1]. I understand that, due to the retrospective nature of the work, it's not possible to reproduce the identical conditions, therefore the word "validation" in the title, should be replaced in order not to mislead the reader. 

Again, the same abbreviations are used to description of different features. As I've pointed in last review in one part of manuscript lLND means limited template, but in other means <10 excised LN's. 

[1] Collins GS, Reitsma JB, Altman DG, Moons KG. Transparent reporting of a multivariable prediction model for individual prognosis or diagnosis (TRIPOD): the TRIPOD statement. BMJ. 2015 Jan 7;350:g7594. doi: 10.1136/bmj.g7594. PMID: 25569120.

Author Response

Dear Editor,

Thank you for your comments.

We should agree that the main question, which was pointed by the reviewer, how reliable is analysis 70% based on the limited number of LN removed. In our opinion, the real-life reflect mixed lPLND and ePLND. If our results are similar to detected in cohorts with more extended PLND, we can suggest using them in daily practice.

We agree with the reviewer arguments regarding the word “validation” and removed them from the title (line 1) and text (lines 15, 59, 105, 148, 202, 260)

We made changes to clarify the extent of PLND (<10> lymph node removed) in the beginning of material and methods (lines 68-72; 81-88) and additionally we made some changes in a better description of statistical analysis (lines 119-128)

Sincerely,
Zilvinas Venclovas and Daimantas Milonas​